# Two Conserved Amino Acids Characterized in the Island Domain Are Essential for the Biological Functions of Brassinolide Receptors

**DOI:** 10.3390/ijms231911454

**Published:** 2022-09-28

**Authors:** Wenjuan Li, Jiaojiao Zhang, Xiaoyi Tian, Hui Liu, Khawar Ali, Qunwei Bai, Bowen Zheng, Guang Wu, Hongyan Ren

**Affiliations:** College of Life Sciences, Shaanxi Normal University, Xi’an 710119, China

**Keywords:** brassinosteroids, BRI1, receptor function, positive selection

## Abstract

Brassinosteroids (BRs) play important roles in plant growth and development, and BR perception is the pivotal process required to trigger BR signaling. In angiosperms, BR insensitive 1 (BRI1) is the essential BR receptor, because its mutants exhibit an extremely dwarf phenotype in Arabidopsis. Two other BR receptors, BRI1-like 1 (BRL1) and BRI1-like 3 (BRL3), are shown to be not indispensable. All BR receptors require an island domain (ID) responsible for BR perception. However, the biological functional significance of residues in the ID remains unknown. Based on the crystal structure and sequence alignments analysis of BR receptors, we identified two residues 597 and 599 of AtBRI1 that were highly conserved within a BR receptor but diversified among different BR receptors. Both of these residues are tyrosine in BRI1, while BRL1/BRL3 fixes two phenylalanines. The experimental findings revealed that, except BRI1Y597F and BRI1Y599F, substitutions of residues 597 and 599 with the remaining 18 amino acids differently impaired BR signaling and, surprisingly, BRI1Y599F showed a weaker phenotype than BRI1Y599 did, implying that these residues were the key sites to differentiate BR receptors from a non-BR receptor, and the essential BR receptor BRI1 from BRL1/3, which possibly results from positive selection via gain of function during evolution.

## 1. Introduction

Brassinosteroids (BRs) are important phytosterol hormones that regulate numerous biological functions of plant growth and development, including photomorphogenesis, cell elongation and division, vascular differentiation, pollen development, plant senescence, and stress resistance [1,2,3,4,5]. Plants that lack BR signaling exhibit retarded growth with dwarf phenotypes and male infertility and are unable to complete their life cycles. BRs are perceived by BR receptors at the cell surface to start a signaling cascade. In *Arabidopsis thaliana*, BRI1 is the most essential BR receptor due to the extremely dwarf phenotype of the null *bri1* mutant, similar to the phenotype observed in the strongest BR biosynthetic mutants [3,6,7,8]. BRI1 belongs to group X of leucine-rich repeat receptor-like kinases (LRR-RLKs), a large cell surface receptor family in terrestrial plants. The typical structure of BRI1 consists of an extracellular domain made of 25 leucine-rich repeats (LRR), a single-pass transmembrane segment (TM), and an intracellular cytoplasmic region containing a Ser/Thr kinase domain (KD) and a c-terminal domain that inhibits the activity of the intracellular kinase. Near the plasma membrane, an island domain (ID) with 70 residues is embedded within the 21st–22nd LRRs, and together they form a BL binding domain (BD). Mutations in the ID directly affect the BR-binding ability of BRI1 [9,10]. Canonical BR signaling begins with the binding of BR to the cell membrane receptor BRI1 [9,10,11]. After BR perception, BRI1 is activated through trans- and auto-phosphorylation with its co-receptor BAK1, which leads to the release of its inhibitor BKI1 (BRI1 Kinase Inhibitor 1) [12,13]. After a series of phosphorylation and dephosphorylation downstream events, the transcription factors brassinosteroid-EMS-suppressor 1 (BES1) and brassinazole resistant 1 (BZR1) are activated through dephosphorylation and eventually moved into the nucleus, where they regulate the expression of thousands of BR-responsive genes; therefore, the dephosphorylation of BES1/BZR1 is a hallmark of the activated BR signal pathway [14,15,16,17,18].

Brassinolide (BL) is the most active form of BRs. The co-crystal structure of BRI1 and BL shows that one molecule of BL can bind to the proximity of the ID in a single BRI1 monomer (Appendix A). The A-D rings of BL bind to a hydrophobic surface between LRRs 23 and 25 and the ID of BRI1, while the alkyl chain is positioned in a small pocket formed by LRRs 21–22 and the ID [19,20]. The interactions between BL and the residues of the BD are critical for binding BL. Mutations in the residues of ID and nearby LRRs greatly affect BL binding, such as loss-of-function alleles *bri1-9* (S662F) [21] and *bri1-113* (G611E) [22] which may interfere with the folding of the ID. Many studies have shown that the BRI1 ID residues involved in BL perception include Y597, Y599, H645, and S647 [10,23]. When BL binds to BRI1, the fused ring moiety of BL is located in most of the surface groove. The edge of the A-ring is in contact with the concave surface, the B-ring is tightly stacked with Y642, and the other two rings are sandwiched between F681 and Y599. The hydroxyl group of Y599 interacts with the carbonyl oxygen at C6 of BL to form a hydrogen bond. The alkyl chain of BL is completely embedded in the small pocket formed by LRR 21 and 22 and two loops connecting the ID and the LRR core. A hydrogen bond is established between the hydroxyl on C23 and S647 on BRI1, and two water-mediated hydrogen bonds are established with carbonyl oxygen atoms of H645 and the hydroxyl group of Y597 [19,23]. The activation of BRI1 by BL is achieved by the induced-fit mechanism [10,19,20]. BL can cause the rearrangement of two loops connecting the ID and LRR structures and lead to the formation of a more obvious surface groove at the BL binding region that participates in the BRI1–BAK1 interaction [19]. The gain-of-function allele, *sud1*, produces an ordered loop even in the absence of steroid ligands, while other mutations affecting this twisted loop lead to the inactivation of BRI1 receptors [21,24,25].

There are three BRI1-like receptors in *Arabidopsis thaliana*: BRI1-like 1 (BRL1), BRI1-like 2 (BRL2), and BRI1-like3 (BRL3), all of which are known to specifically express in vascular tissues, albeit with some minor differences [26,27,28]. BRL1 and BRL3 have been reported to promote xylem differentiation, while BRL2 is necessary for maintaining leaf provascular differentiation [26,27,29]. BRL1 and BRL3 have the closest homology with 80% sequence identity, indicating that BRL1 and BRL3 have diverged recently and may have similar functions, therefore, we used BRL1/3 in our study to represent both genes. The sequence identities of BRL1/3 and BRL2 with BRI1 are 43% and 41%, respectively [26,27,30,31]. Although these sequence identities are almost the same, the functions of BRL1/3 and BRL2 are uniquely different from each other. BRL1/3 can rescue the phenotype of *bri1* mutants when overexpressed under the BRI1 promoter, while BRL2 cannot. Consistently, the biochemical analysis found that both BRL1 and BRL3 can bind BL, while BRL2 showed no specific BL binding activity [26,32]. These results may seem unexpected due to highly conserved residues in the ID and their surrounding residues. The crystal structure of BRL1 in complex with BL shows that although the extracellular domain of BRI1 and BRL1 are highly conserved, their binding sites with BL are slightly different, and these subtle differences may have major implications for their ability to recognize BL. In BRL1, Phe586 and Tyr627 interacted with the ID side of BL, while Gln666 established Van Der Walls contact with the LRRs side of BL. The distal side chain of BL is anchored to the hydrophobic cavity formed by Ile527 and Trp551 of the LRRs side and Phe584, Leu601, and Thr631 of the island side. The C23 hydroxyl of BRL1 forms hydrogen bonds similar to those of BRL1, but it also binds to the backbone nitrogen and oxygen of Met632 (Appendix A). Many structural studies have solved the crystal structures of BRI1 and BRL1 and provide insights into the recognition of steroid hormones by these receptors, but residues responsible for the biological function of BR receptors, and the potential diversification mechanisms of different receptors binding to BL are not clear. Here, we identified two important residues highly conserved within a specific BR receptor but diversified among different types of BR receptors. In angiosperms, around 90% of BRI1 from different species evolved to have a tyrosine at position 597, and only 10% still keep the phenylalanine which is the only fixed amino acid for all the BRL1 at the same position (584 of AtBRL1), where BRL2 favors four different types of amino acids. In addition, all the BRI1 chose a tyrosine at position 599, but all the BRL1 and BRL2 selected phenylalanine. To investigate the biological functional significance of these differentiated residues, we mutated the residues 597 and 599 of AtBRI1 to the other 19 possible amino acids, respectively, and found that except BRI1Y597F and BRI1Y599F, all the mutated BRI1 differently reduced or even lost their BR signaling, and mutation analysis of these residues of AtBRL1 also proved the same. Surprisingly, we also discovered that AtBRI1Y599F gave a weaker phenotype than AtBRI1Y599 did, suggesting diversification between the essential BR receptor BRI1 and BRL1/3 for BR perception, which possibly facilitates the dominance of BRI1 in angiosperm compared to BRL1/3.

## 2. Results

### 2.1. Tyrosine597 and Tyrosine599 in the BRI1 Island Domain Are Fixed during Plant Evolution

To better understand the phylogenic relationships of BR receptors, a Bayesian phylogeny was constructed and showed that BRI1 grouped with BRL1/3 but not with BRL2 (Figure 1A). Next, we searched for the orthologs of BRI1, BRL1/3, and BRL2 in the databases, and obtained 331 full-length protein sequences (Appendix A). As ID is the most important region for the BR-binding function of BR receptors [9,19], we then used the ID of these sequences to construct the phylogenetic tree, and found that the IDs of BRI1 grouped into a clad and formed a subgroup together with a clad of BRL1/3 IDs, distinct from BRL2. On the contrary, IDs of BRL2 were highly dispersed, implying that the ID of BRL2 was possibly not as requisite as that of BRI1 and BRL1/3 for its BL binding function, thereby the conserved residues in the ID are critical to identifying BR receptors from non-BR receptors (Appendix A). Crystal structure analysis demonstrated that conserved residues in the ID are responsible for BR perception (Appendix A). We constructed a WebLOGO to evaluate the divergence of ID sequences among BRI1, BRL1/3, and BRL2. The results showed that the conserved residues were scattered along the ID, most of which were shared among BRI1, BRL1/3, and BRL2, while a few amino acids, tyrosine (residue 597 of AtBRI1) and tyrosine (residue 599 of AtBRI1), for instance, were only conserved within the BRI1 group (Figure 1B). Our data also showed that 90% of the residues at position 597 were tyrosine (Try/Y), while the remaining 10% were phenylalanine (Phe/F). All the amino acids of the BRL1/3 group at the corresponding position (584 of AtBRL1) were phenylalanine (Phe/F), while BRL2 possessed various kinds of amino acids (Figure 1C and Appendix A). Residue 599 of BRI1 was more conserved compared to residue 597 among these receptors, as all BRI1 from different species kept a tyrosine for residue 599, and all BRL1/3 and BRL2 had phenylalanine (Figure 1D). These results suggest that during the course of evolution, residue 599 may play a more critical role than residue 597 for BR recognition, and the subtle differences between BRI1 and BRL1/3 in BR perception may also be related to these two residues.

### 2.2. Tyrosine597 and Phenylalanine597 Differentiate BR Receptors from Non-BR Receptor

Next, we argued that if these two tyrosines are fixed for BRI1, tyrosine at these two residues could play some role in BR binding. To test this hypothesis, we substituted Tyrosine597 in AtBRI1 with the remaining 19 amino acids, respectively, fused them with a GFP tag, and then expressed them under a BRI1 promoter in a weak *bri1* mutant *bri1-301*. Interestingly, we found that only transgenic plants expressing BRI1 and BRI1Y597F showed phenotypes of obviously elongated petioles and enlarged rosette leaves, whereas transgenic plants of BRI1Y597M, BRI1Y597A, BRI1Y597I, and BRI1Y597L, where the corresponding BRI1 residues were replaced with that of BRL2, presented much weaker phenotypes. To our surprise, among all amino acids, BRI1Y597W and BRI1Y597P could partially rescue *bri1-301*, suggesting that during evolution, both Tyrosine597 and Phenylalanine597 were selected for BR perception, while BRL2 evolved with amino acids even weaker than tryptophane (Trp/W) and proline (Pro/P) (Figure 2A). Statistical analysis also showed significant differences of petiole lengths between BRI1Y597F and BRI1Y597M and between BRI1Y597F and BRI1Y597W, but not between Col-0 and BRI1Y597F (Figure 2B). The protein levels of ectopically expressed receptors in transgenic plants were almost equal, indicating that these phenotypes were unrelated to the amount of protein (Figure 2C). Accordingly, we proposed that Tyrosine597 and Phenylalanine597 are commensurate with each other and are vital in binding BRs. To further investigate this theory, we also changed the corresponding amino acid residue in AtBRL1 (584 of AtBRL1) into the other amino acids, respectively. Similarly, both BRL1Y584 and BRL1F584Y complemented the dwarf phenotypes of *bri1-301*, which is consistent with our previous findings, while the rest of the substituted amino acids differently impaired the function of BRL1 (Figure 3A). The protein expressions were examined, and statistical analysis also supported the same results (Figure 3B,C). Taken together, the above findings suggest that Tyrosine597 and Phenylalanine597 are functionally conserved, distinguishing BR receptors from the non-BR receptor.

### 2.3. Tyrosine599 Specifies the Essential BR Receptor BRI1 from BRL1/3

Considering the fact that all of BRL1/3 and BRL2 keep a phenylalanine, and BRI1 prefer a tyrosine, residue 599 of AtBRI1 is more conserved than residue 597. Thus, we further speculated that Tyrosine599 would show more requisite and specific features. To test this hypothesis, we applied the same phenotype complementation method to evaluate the substitutions of residue 599 with different amino acids. The analysis suggested that BRI1Y599F rescued the weak mutant *bri1-301*, yet the phenotypes of its transgenic plants were much weaker than BRI1Y599, with less elongated petioles and rounder leaves. All the other substituted 18 amino acids differently impaired the function of BRI1, indicating that any mutations of Tyrosine599 and Phenylalanine599 would weaken the function of BR signaling (Figure 4A). The same conclusion was derived from statistical analysis of petiole length and from protein expression levels (Figure 4B,C). Furthermore, we substituted the phenylalanine of the corresponding residue of AtBRL1 (586 of AtBRL1) with different amino acids and found that, apart from BRL1F586 and BRL1F586Y, the transgenic plants of which share similar phenotypes, the substitutions showed a much weaker phenotype in the *bri1-301* background, raising the possibility that Tyrosine599 also promoted the perception of BRs by intramolecular interaction (Figure 5A). The results were further confirmed with statistical analysis, and protein expression levels of ectopic proteins were almost equal (Figure 5B,C). In conclusion, Tyrosine599 enhanced BR signaling for BRI1, thus differentiating the essential BR receptor BRI1 from other BR receptors.

### 2.4. BR Receptors with Tyrosine597 or Tyrosine599 Showing the Highest Biochemical and Physiological Efficacy in BR Signaling

During evolution, Tyrosine597 and Tyrosine599 change BR signaling in different manners, so we asked whether these changes also happen at the biochemical and physiological levels. To answer this question, we examined the dephosphorylation vs. phosphorylation status of transcription factor BES1, since BES1 dephosphorylation is a hallmark of BR signaling. When Col-0 is treated with 24-epibrassinolide (24-eBL), dephosphorylated BES1 greatly accumulates, while phosphorylated BES1, on the other hand, significantly decreases. We found that, compared to the wild type, BRI1Y597F increased the ratio of dephosphorylated BES1 as BRI1Y597 did, but two substitutions BRI1Y597L and BRI1Y597A showed less dephosphorylated BES1 activity, implying possible impaired BR signaling. Similarly, BRI1Y597W and BRI1Y597P also showed slightly increased accumulated dephosphorylated BES1 activity similar to Col-0, which is consistent with the partially rescued phenotypes of the transgenic plants. All these results proved that different amino acids of residue 597 that is responsible for BR perception result in different intensities for BR signaling (Figure 6A). Next, we asked whether tyrosine can also improve BR signaling at the corresponding position of BRL1. Interestingly, BES1 phosphorylation analysis showed that only tyrosine and phenylalanine of residue 584 of AtBRL1 (BRL1 and BRL1F584Y) could obviously trigger the BR signaling pathway to dephosphorylate BES1, and others showed much weaker accumulation of dephosphorylated BES1 (Figure 6B). Similar results were observed in a root sensitivity assay to eBL. In the root sensitivity assay, as the concentrations of eBL for root treatment were gradually increased, the root lengths of transgenic plants with a stronger BR receptor would be significantly decreased compared to those with weaker BR receptors or null receptors. Our results demonstrated that of all the substitutions of residue 597 of BRI1, tyrosine and phenylalanine (BRI1Y597 and BRI1Y597F) showed the most sensitivity to eBL treatment, and of all the substitutions in the same residue of BRL1, tyrosine and phenylalanine (BRL1F584 and BRL1F584Y) were the most sensitive to eBL. Statistical calculations also support the same results (Appendix A). Moreover, we also analyzed BES1 dephosphorylation and a root sensitivity assay for Tyrosine599. BES1 dephosphorylation analysis demonstrated that except for BRI1Y599 and BRI1Y599F, much weaker dephosphorylated BES1 was observed in other transgenic plants. Significantly, the accumulation of dephosphorylated BES1 in BRI1Y599F was lower than in BRI1Y599, which is consistent with our findings from transgenic plants where BRI1Y599F transgenic plants had a weaker phenotype than BRI1Y599, leading to the conclusion that Tyrosine599 indeed promoted BR signaling for BRI1, which may enhance the efficacy of BRI1 in angiosperm (Figure 6C). Similarly, we also learned that of all the substitutions at the corresponding position of BRL1 (586 of AtBRI1), BRL1F586 and BRL1F586Y both showed the accumulation of dephosphorylated BES1, but others showed much lower dephosphorylated BES1 accumulation, suggesting the possibility of cooperation between residue 599 and other residues (Figure 6D). The root growth inhibition assay also indicated that the roots of both BRI1F599 and BRI1F599Y are sensitive to BR treatment, and BRL1F586 and BRL1F586Y have the most sensitive roots of all. Similar results were inferred from statistical analysis as well (Appendix A). In conclusion, our findings prove that both Tyrosine597 and Tyrosine599 could determine the high efficacy of BRI1-mediated BR signaling, which maybe provide a basis for BRI1 prevalence as an essential BR receptor.

## 3. Discussion

As we know, amino acids are the building blocks for the 3D protein conformations that virtually determine the unique biological function of numerous enzymes, peptides, and other proteins, including interaction with a small molecule or binding a ligand [33,34,35]. Within a protein family, to maintain the conserved structure and shared functional features, many substitutions of one residue or segments of one protein with its homolog cause no phenotype change [36]. However, once a protein or an organism is subjected to strong positive selection or is neo-functionalized, substitutions of a few amino acids in a critical residue of a protein may result in conformational changes in the protein molecule, thus allowing the protein or an organism to acquire a new feature—gain-of-function [37,38,39]. For instance, a single amino acid substation in a Ser/Thr protein kinase glycogen synthase kinase 3 (STKc_GSK3) could lead to a stabler response of STKc_GSK3 to BRs, contributing to the formation of the round grain of wheat and further the origin of Triticum sphaerococcum [40]. As for the LRR-RLK family of Arabidopsis, different types of molecules such as brassinosteroids or phytosulfokin are used as their ligands, implying that conformational differences in the binding sites of LRR-RLKs are required—a series of mutations with gain-of-function are needed.

The radiation of land plants is accompanied by the expansion of the LRR-RLKs family. According to a previous study, BR receptors, including BRI1, BRL1/3, BRL2, and Excess Microsporocytes 1 (EMS1) and Nematode-Induced LRR-RLK 1 (NILR1) share the common downstream pathway, suggesting that a pan-brassinosteroid signaling pathway exists in land plants, mediated by a different receptor like kinase. A close evolutionary relationship between EMS1 and BR receptors indicates that they may have evolved from a common ancestor [18,41,42]. The phylogenetic analysis illustrated that BRI1 falls into the same clad with BRL1/3 but not with BRL2, suggesting that BR perception is a vital requirement for BR receptor evolution (Figure 1A). The divergence of receptors sharing common cascades mainly occurs on the extracellular domain of these receptors, accounting for exchangeable KDs of proteins in these receptors. Of the extracellular domain, island domains (IDs), essential for BR binding, are highly diversified. Here, we showed the significant evolutionary importance of two pivotal residues in the ID for BR perception: (i) of all 20 amino acids, BRI1 fixed Tyrosine597 and Tyrosine599, and correspondingly, BRL1 fixed Phenylalanine584 and Phenylalanine586 to perceive BRs, whereas BRL2 opted for various amino acids at the same residue; (ii) both Tyrosine597 and Phenylalanine597 showed the strongest BR signaling either in BRI1 or in BRL1, which was further proved by BES1 phosphorylation analysis and eBL treatment analysis, indicating that residue 597 of BRI1 differentiated BR receptors from a non-BR receptor through merely two amino acids, i.e., tyrosine and phenylalanine; (iii) tyrosine, of all amino acids at residue 599 of BRI1, had the strongest BR signaling output, and both tyrosine and phenylalanine in the same residue of BRL1 presented the strongest phenotypes, implying that Tyrosine599, through intramolecular interaction, determined the dominance of BRI1 in the angiosperm, other than BRL1. A possible binding analysis can be implemented further to verify these results.

One possible hypothesis could be proposed that BRI1 acquired these two amino acids through a series of gain-of-function mutations. Functional residue migration refers to the substitutions of a residue that can be fixed if they can take over the roles of old amino acids under the constraints of selection [43,44]. In general, conserved residues are related to important functional roles; thereby a fixed substitution in a conserved residue always indicates functional residue migration or gain-of-function. With two gene-duplication events, a common ancestor of BR receptors diversified with BRL2 first, and then BRI1 separated with BRL1/3 [42,45,46,47]. Obviously, BRL2 possesses various amino acids at residue 597 (of AtBII1), suggesting that functional residue migration might have caused this phenomenon. Furthermore, BRI1 and BRL1 show more conserved residues, meaning that tyrosine and phenylalanine are irreplaceable for BR perception, the function of which was probably gained by the common ancestor of BR receptors for the same residue. After the second duplication, since the essential BR receptor BRI1 co-evolved with its most active BR—BL, BRI1 acquired a Tyrosine599 stronger in activity than Phenylalanine586 of BRL1, and this may enable the high-efficacy and broad expression for BRI1. Our study tried out all the possible amino acids and rebuilt the evolutionary paths for two key binding residues of BR receptors, providing new insights into the molecular evolution.

## 4. Materials and Methods

### 4.1. Plant Materials

The *Arabidopsis thaliana* Columbia (Col-0) ecotype obtained from “the Arabidopsis Information Resource” (https://www.arabidopsis.org/, accessed on 1 July 2012) was used as a wild type in this study, and the BRI1 weak mutant *bri1-301* in the Col-0 background was selected to analyze the function of mutated receptors. Seeds of Arabidopsis were surface-sterilized with 75% (*v*/*v*) ethanol for 5 min, followed by 5% (*v*/*v*) NaClO for 1 min, and then grown on a ½ Murashige and skoog (1/2 MS) medium [32]. One week after germination, the seedlings were transferred to soil and cultured in a greenhouse under a 16 h light/8 h dark cycle at 19–21 °C.

### 4.2. Generation of Transgenic Plants

A Ti plasmid pCHF3 with an AtBRI1 promoter and a GFP tag was used to generate transgenic Arabidopsis plants in this study. The full-length CDSs of AtBRI1 and AtBRL1 were amplified from the cDNA of Arabidopsis thaliana, and mutations of AtBRI1 or AtBRL1 were generated from PCR products using specific primer pairs. Then, the purified PCR products were cloned into pCHF3 via a homologous recombination technology with a ClonExpress II One Step Cloning Kit (Vazyme, Nanjing, China). All the constructs were verified by sequencing to confirm there were no PCR-induced mutations. The resulting constructs were then introduced into a Agrobacterium tumefaciens strain GV3101 by the freeze-thaw method and transformed into *bri1-301* using the floral dip method [48]. All the primers used for PCR amplification are listed in Appendix A.

### 4.3. eBL Treatment of Root

Seeds from two independent transgenic lines were surface-sterilized and sown on 1/2 MS media supplemented with 24-epibrassinolide (24-eBL, Sigma, St. Louis, MO, USA) at the concentrations of 0, 10, and 100 nM. After being incubated at 4 °C for 2 days, the seeds were grown vertically under a 16 h light/8 h dark cycle at 19–21 °C for 7 days. The seedlings with uniform growth were then photographed. Their root lengths were measured with Image J and analyzed with GraphPad Prism 8.0 (http://www.graphpad.com, accessed on 17 January 2020, GraphPad Software, San Diego, CA, USA).

### 4.4. Immunoblot Analysis

Two-week-old transgenic seedlings were collected and ground into a fine powder in liquid nitrogen. Total protein extracts were extracted by adding 2× of SDS-PAGE sample buffer. After being incubated at 100 °C for 10 min, the mixture was centrifuge at 14,000 rpm for 10 min at 4 °C, and the supernatants were collected. Total protein extracts in the supernatants were then separated on a 10% SDS-PAGE gel and transferred into a PVDF membrane (Pall, Beijing, China). An anti-GFP antibody (1:1000 dilution, Transgen, Beijing, China) was used to detect GFP fusion proteins, and an anti-BES1 antibody (1:3000) dilution (provided by Li Jia, Lanzhou University, Lanzhou, China) was used to quantify the phosphorylation vs. dephosphorylation status of BES1. Actin, detected by anti-Actin antibodies (1:1000; Abmart, Shanghai, China), served as the loading control for both protein expression analysis and BES1 phosphorylation analysis.

### 4.5. Phylogenetic Analysis

Full-length protein sequences of AtBRI1, AtBRL1/3, and AtBRL2 were retrieved from “the Arabidopsis Information Resource” (https://www.arabidopsis.org/, accessed on 17 January 2020). The retrieved sequences were used as query sequences to pBLAST against phytozome v.13 (https://phytozome-next.jgi.doe.gov/, accessed on 18 January 2020), China National GeneBank DataBase (https://db.cngb.org/blast/blast/tblastn/, accessed on 18 January 2020), and Ambrorella (http://www.amborella.org, accessed on 18 January 2020) to obtain homologs of these genes. Orthologs of each gene were selected from homologs by pBLAST against the protein database of *Arabidopsis Thaliana* with the best e-value. Multisequence alignments were then performed with MAFFT 7.0 (https://www.ebi.ac.uk/Tools/msa/mafft/, accessed on 20 January 2020). Sequence logo was performed using Weblogo online (https://weblogo.berkeley.edu/logo.cgi, accessed on 28 January 2020). Bayesian phylogenic trees were constructed using IQ-TREE (http://iqtree.cibiv.univie.ac.at/, accessed on 5 June 2020) and visualized with FigTree v 1.4.4 (http://tree.bio.ed.ac.uk/software/figtree/, accessed on 20 June 2020).

### 4.6. Structure Analysis

The X-ray structures of BRI1 (3RGX) and BRL1 (4J0M) were obtained from the Protein Database Bank (https://www.rcsb.org, accessed on 22 January 2020) and then visualized and labelled using PyMOL Molecular Graphic systems (https://pymol.org, accessed on 22 January 2020).

### 4.7. Statistics Analysis for Petiole Lengths

25-day-old seedlings were photographed and measured with Image J with 15 replicates for each gene. The data were then analyzed using one-way analysis of variance (ANOVA) with GraphPad Prism 8.0.

## Figures and Tables

**Figure 1 ijms-23-11454-f001:**
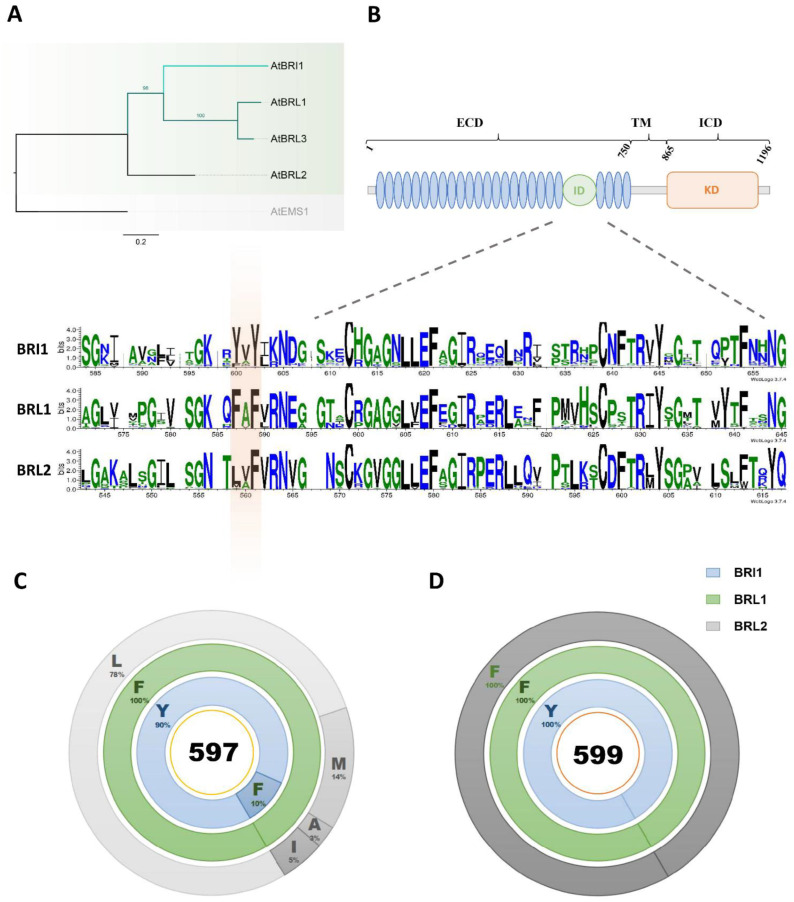
Two residues are fixed in the ID of BR receptors for BR perception. (**A**) Phylogenetic relationship of BR receptors—BRI1 and BRL1/3, and non-BR receptor BRL2. The phylogeny was generated by the IQ-TREE web server, and the confident values were estimated using the SH-aLRT branch test with 1000 replicates. EMS1 served as an outgroup. Gray branches indicate non-BR receptors, while green and blue branches highlight the BR receptors. (**B**) Schematic representation of a typical BR receptor with an extracellular domain (ECD), a transmembrane domain (TM), and an intracellular domain (ICD), and the Sequence logos’ analysis presenting the conservations of the ID among BR receptors. Colors of symbols in Sequence logos show different hydrophobicity, while the symbols’ stack indicates the conservation of a residue in the sequence. Two conserved residues (597 and 599 of AtBRI1) are highlighted in the orange background. (**C**) Donut charts of the amino acid probability of residue 597 in BRI1 (blue donut), BRL1/3 (green donut), and BRL2 (grey donut). Different letters with percentages below show different amino acids with their probabilities in position 597 (AtBRI1). (**D**) Donut charts of the amino acid probability of residue 599 in BRI1 (blue donut), BRL1/3 (green donut), and BRL2 (grey donut). Different letters with percentages below show different amino acids with their probabilities in position 599 (AtBRI1).

**Figure 2 ijms-23-11454-f002:**
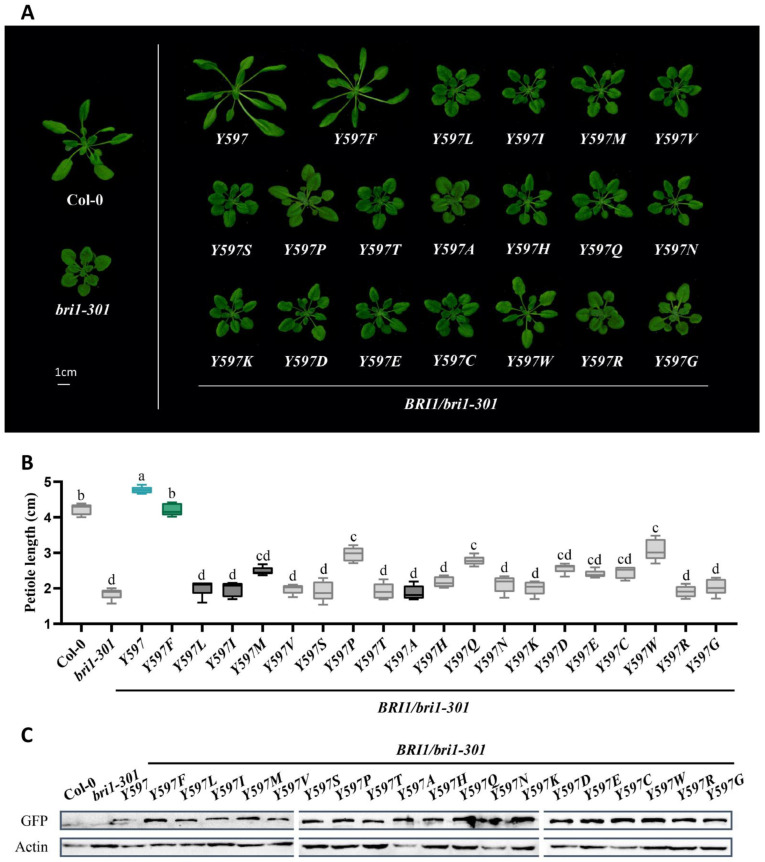
The tyrosine and phenylalanine of residue 597 distinguish the BR receptors from the non-BR receptor. (**A**) Phenotypes of wild-type Col-0, *bri1-301*, and transgenic plants with mutated receptors expressed in *bri1-301*. Four-week-old seedlings were presented. Scale bar = 1 cm. (**B**) Petiole length comparison of transgenic plants. The blue box highlights Tyrosine597 of BRI1, the green marks phenylalanine fixed in BRL1/3, the black represents the amino acids from BRL2, and the grey indicates the remaining amino acids. The results are represented as the means ± SD. Different letters indicate significant differences (*n* = 15, *p* < 0.0001, one-way ANOVA with a Tukey’s test). (**C**) The protein expression level of mutated receptors with a GFP tag in transgenic plants. An anti-GFP antibody was used to quantify the expression level.

**Figure 3 ijms-23-11454-f003:**
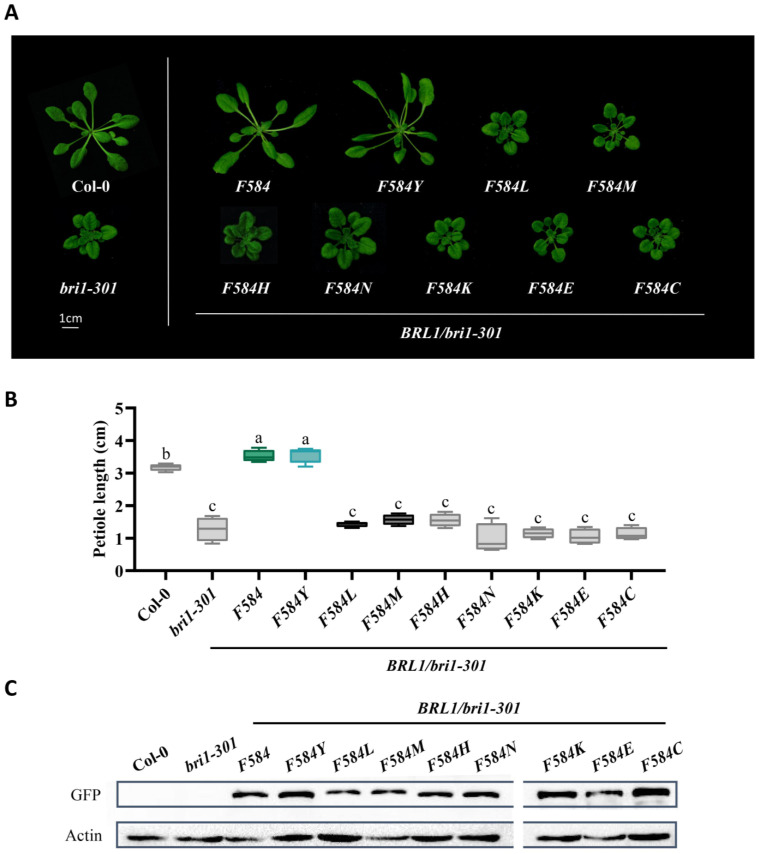
The tyrosine and phenylalanine of residue 584 of BRL1 (corresponding to residue 597 of AtBRI1) conferring BR receptors the highest efficacy. (**A**) Phenotypes of 4-week-old seedlings of wild-type Col-0, *bri1-301*, and transgenic plants with mutated receptors expressed in *bri1-301*. Scale bar = 1 cm. (**B**) Measurements of petiole length of the transgenic plants shown in (**A**). The blue box highlights tyrosine fixed in BRI1, the green marks the Phenylalanine584 of BRL1/3, the black represents the amino acids fixed in BRL2, and the grey indicates the remaining amino acids. Different letters indicate significant differences (*n* = 15, *p* < 0.0001, one-way ANOVA with a Tukey’s test). (**C**) Protein expression level of mutated receptors.

**Figure 4 ijms-23-11454-f004:**
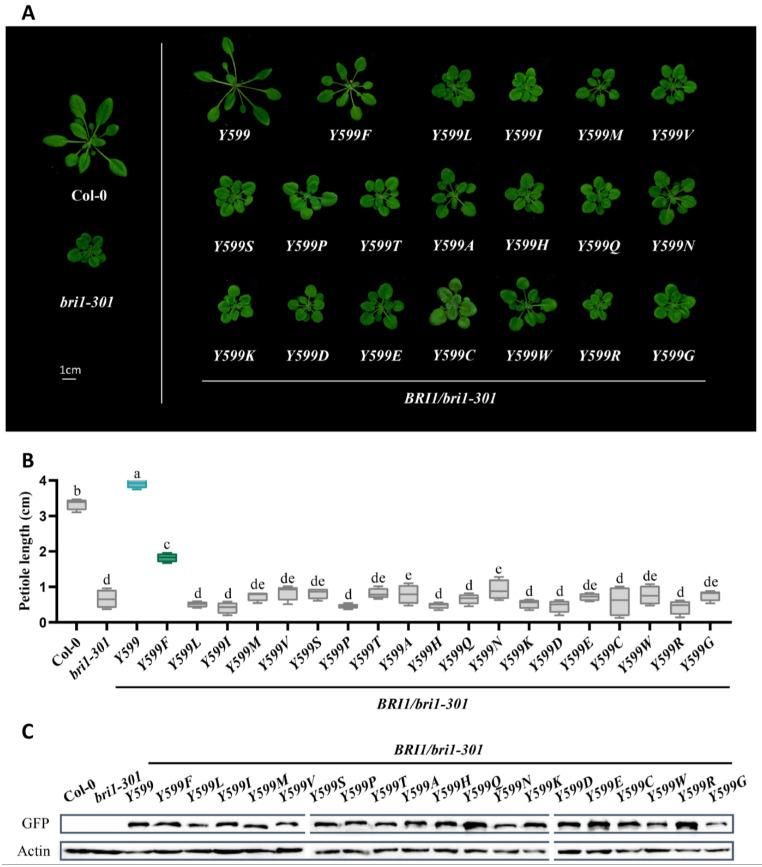
The tyrosine of residue 599 specifies the essential BR receptor BRI1 from BRL1/3. (**A**) Phenotypic analysis of 4-week-old seedlings of wild-type Col-0, *bri1-301*, and transgenic plants with mutated receptors expressed in *bri1-301*. Scale bar = 1 cm. (**B**) Analysis of petiole length represented as the means ± SD. The blue box highlights Tyrosine599, the green indicates phenylalanine fixed in BRL1/3, and the grey represents the remaining amino acids. Different letters indicate significant differences (*n* = 15, *p* < 0.0001, one-way ANOVA with a Tukey’s test). (**C**) Protein expression comparison of mutated receptors tagged with GFP in transgenic plants. An anti-GFP antibody was used to quantify the expression level.

**Figure 5 ijms-23-11454-f005:**
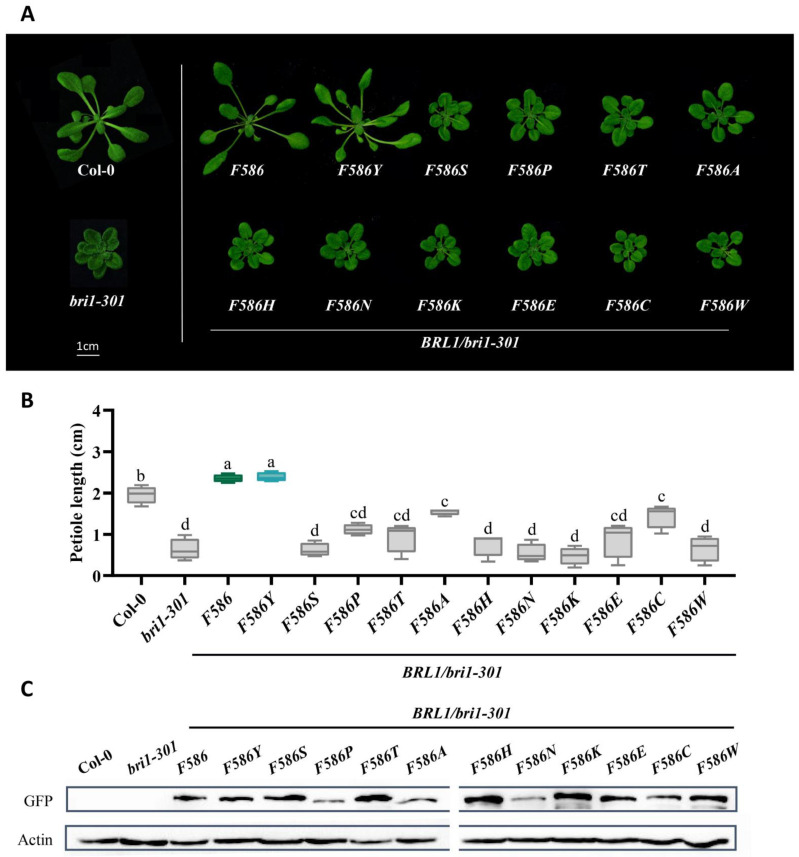
Both tyrosine and phenylalanine at position 586 of AtBRL1 (corresponding to residue 599 of AtBRI1) presented the strongest phenotypes. (**A**) Phenotypes of wild-type Col-0, *bri1-301*, and transgenic plants with mutated receptors expressed in *bri1-301*. Scale bar = 1 cm. (**B**) Petiole length analysis of the plants shown in (**A**). The results are represented as the means ± SD. The blue box highlights tyrosine fixed in BRI1, the green indicates Phenylalanine586 of BRL1/3, and the grey represents the remaining amino acids. Different letters indicate significant differences (*n* = 15, *p* < 0.0001, one-way ANOVA with a Tukey’s test). (**C**) Protein expression analysis of GFP-tagged mutated receptors in transgenic plants. An anti-GFP antibody was used to quantify the expression level.

**Figure 6 ijms-23-11454-f006:**
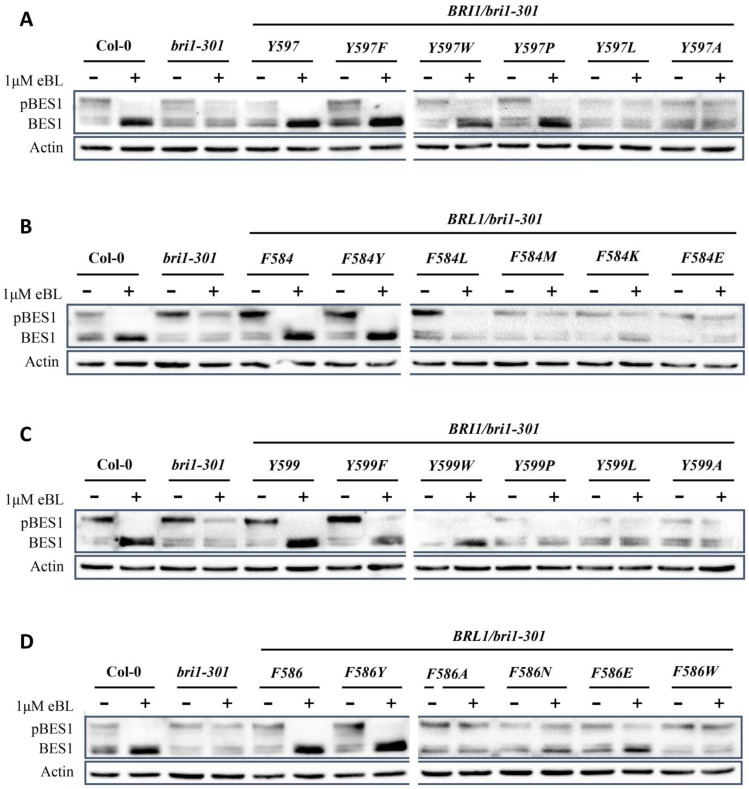
BR receptors with Tyrosine597 or Tyrosine599 showing the highest efficacy to dephosphorylate BES1. (**A**–**D**) Immunoblotting of eBL-induced dephosphorylation of BES1 in Col-0, *bri1-301*, and transgenic plants in a *bri1-301* background. (**A**) Dephosphorylation vs. phosphorylation level of BES1 in transgenic plants expressing AtBRI1 mutated at position 597. (**B**) Dephosphorylation vs. phosphorylation level of BES1 in transgenic plants expressing AtBRL1 mutated at position 584 (corresponding to residue 597 of AtBRI1). (**C**) Dephosphorylation status of BES1 in transgenic plants expressing AtBRI1 mutated at position 599. (**D**) Dephosphorylation analysis of BES1 in transgenic plants expressing AtBRL1 mutated at position 586 (corresponding to residue 599 of AtBRI1). An anti-BES1 antibody was used to detect the phosphorylation status of BES1, and Actin served as a loading control. All experiments were repeated independently three times with similar results.

## Data Availability

The data or material of this study are available from the corresponding author, H.R., upon reasonable request.

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
