# Peer review of "Two Conserved Amino Acids Characterized in the Island Domain Are Essential for the Biological Functions of Brassinolide Receptors"

_ijms, 2022, doi:10.3390/ijms231911454_

Round 1

Reviewer 1 Report

Li’s article revealed a positive selection of Brassinosteroids (BRs) receptor BRI1 and BRL1/3 through carefully designed experiments and clear descriptions, thus I think this article is suitable for IJMS. This article also provided an insight into future studies of BR receptor protein structure and functional mechanism. It is no doubt an interesting article despite of few writing mistakes that will be easily corrected by re-editing, but some thoughts I recommend the authors will consider to describe in the discussion section.

(1)   The author mentioned the BRL1/3 were possibly evolved from non-BR receptor, is there any possible candidates? Because considering the closed relationship between BRL1/3 and BRI1, they may derive from the same ancestor.

(2)   In the discussion section, the authors mentioned BRL1/3 locates in vascular. How about BRI1 and 2? Is signal peptide involved in this phenomenon?

Author Response

Referee: 1

Comments and Suggestions for Authors

Li’s article revealed a positive selection of Brassinosteroids (BRs) receptor BRI1 and BRL1/3 through carefully designed experiments and clear descriptions, thus I think this article is suitable for IJMS. This article also provided an insight into future studies of BR receptor protein structure and functional mechanism. It is no doubt an interesting article despite of few writing mistakes that will be easily corrected by re-editing, but some thoughts I recommend the authors will consider to describe in the discussion section.

Comments:

  1. The author mentioned the BRL1/3 were possibly evolved from non-BR receptor, is there any possible candidates? Because considering the closed relationship between BRL1/3 and BRI1, they may derive from the same ancestor.

Response: We truly appreciate the nice comments from the reviewer and are particularly thankful for the constructive suggestions that make this manuscript much better.

We have added the related content in the discussion. The radiation of land plants is accompanied by the expansion of the LRR-RLKs family. Our study found that the intracellular domain of BRI1, BRL1/3, BRL2 and EMS1 were interchangeable[1, 2], indicating their close relatives. EMS1 is widespread in terrestrial plant, BRL2 is only widely present in vascular plants, while BRI1 and BRL1/BRL3 are present in seed plants[3, 4], these suggest that BRI1 and BRL1/BRL3 may have evolved from BRL2. Thus, the BR receptor family evolved from the common ancestor with EMS1.

  1. In the discussion section, the authors mentioned BRL1/3 locates in vascular. How about BRI1 and 2? Is signal peptide involved in this phenomenon?

Response: Based on previous studies, BRI1 is the most important receptor in angiosperm and widely expresses in the whole plants, and its null mutants (bri1-116) exhibit extremely dwarf phenotype and sterility. On the contrary, BRL2 only expresses in provascular and procambial cells, and its null mutants exhibit defects only in vascular tissues. The expression patterns of BR receptors have been investigated through a series of protomer analyses with a GUS reporter. And we found that the results of GUS staining were consistent with their native expression, indicating that the difference in expression pattern should mainly be caused by the promoter.

We have made a detailed analysis of the sequence and structure of BRL2, and found that the ligand of BRL2 may be a peptide by comparing it with EMS1 (the ligand is a peptide), but the specific information of this peptide needs to be further investigated.

Thanks again for your careful review!

  1. Zheng, B. W.; Bai, Q. W.; Wu, L.; Liu, H.; Liu, Y. P.; Xu, W. J.; Li, G. S.; Ren, H. Y.; She, X. P.; Wu, G., EMS1 and BRI1 control separate biological processes via extracellular domain diversity and intracellular domain conservation. Nat Commun 2019, 10.
  2. Zheng, B.; Xing, K.; Zhang, J.; Liu, H.; Ali, K.; Li, W.; Bai, Q.; Ren, H., Evolutionary Analysis and Functional Identification of Ancient Brassinosteroid Receptors in Ceratopteris richardii. Int J Mol Sci 2022, 23, (12).
  3. Wang, L.; Liu, J.; Shen, Y.; Pu, R.; Hou, M.; Wei, Q.; Zhang, X.; Li, G.; Ren, H.; Wu, G., Brassinosteroids synthesised by CYP85A/A1 but not CYP85A2 function via a BRI1-like receptor but not via BRI1 in Picea abies. J Exp Bot 2021, 72, (5), 1748-1763.
  4. Ferreira-Guerra, M.; Marquès-Bueno, M.; Mora-García, S.; Caño-Delgado, A. I., Delving into the evolutionary origin of steroid sensing in plants. Current Opinion in Plant Biology 2020, 57, 87-95.

Reviewer 2 Report

This may be interesting, but some important points need to be resolved. Importantly, a study must provide a critical analysis of the data. In other words, you must assess whether specific data published really stand up to scientific scrutiny. In order to achieve the above, you must clearly define your specific aims and objectives. So in your study you must develop a critical appraisal of the state of the art. This is an essential element of any article. There are important scientific questions (both conceptual and methodological) which need to be addressed with the primary studies. A study must highlight this. The introduction, which is written in clear language, covers a number of relevant issues. Information are noteworthy, and not are correct supported by similar results from the specialty (see WOS: 000354914500007, WOS: 000346138400005, WOS: 000339050700030, WOS: 000327818000032, WOS: 000327816300004, WOS: 000318221200014). Try to rewrite the abstract and conclusions, I also recommend the nuance of the introduction, the way of working is not very well explained, the procedure is tedious and unsustainable. For this reason, I recommend that the authors try to use more sustainable methodologies, the interpretation of the results can be improved/ reformulated,

Author Response

Referee: 2

Comments and Suggestions for Authors

This may be interesting, but some important points need to be resolved. Importantly, a study must provide a critical analysis of the data. In other words, you must assess whether specific data published really stand up to scientific scrutiny. In order to achieve the above, you must clearly define your specific aims and objectives. So in your study you must develop a critical appraisal of the state of the art. This is an essential element of any article. There are important scientific questions (both conceptual and methodological) which need to be addressed with the primary studies. A study must highlight this. The introduction, which is written in clear language, covers a number of relevant issues. Information are noteworthy, and not are correct supported by similar results from the specialty (see WOS: 000354914500007, WOS: 000346138400005, WOS: 000339050700030, WOS: 000327818000032, WOS: 000327816300004, WOS: 000318221200014). Try to rewrite the abstract and conclusions, I also recommend the nuance of the introduction, the way of working is not very well explained, the procedure is tedious and unsustainable. For this reason, I recommend that the authors try to use more sustainable methodologies, the interpretation of the results can be improved/ reformulated.

Response: Thanks for your careful review!

In this paper, we used genetics, physiological and bioinformatical analysis to experimentally study molecular evolution in vivo. From our study, we found two residues critical in the adaptation of BRI1/BRLs-BR signaling. In residue 597 of AtBRI1, only Tyrosine (of BRI1) and Phenylalanine (of BRL1/3) give the strongest phenotype of BR signaling among all the transgenic plants, which means these two amino acids are the best among all 20 kinds of amino acids, and were specifically fixed for the BRI1/BRLs-BR signaling. As for residue 599 of AtBRI1, Tyrosine (of BRI1) and Phenylalanine (of BRLs) are also the strongest than the remaining amino acids. Notably, Tyrosine showed a stronger phenotype than Phenylalanine in the AtBRI1 background, indicating the differentiation between essential BR receptor BRI1 and BRL1/3. We have carefully checked and sorted out the logic and language of the paper, and made some revisions.

Thanks again for your careful review making this manuscript much better.

Round 2

Reviewer 2 Report

This manuscript entitled "Two conserved amino acids characterized in the island domain are essential for the biological functions of brassinolide receptors" could be good for publication in IJMS (ISSN 1422-0067).